# Efficient Photovoltaic Unit for Power Delivering to Stand-Alone Direct Current Buildings Using Artificial Intelligence Approach Based MPP Tracker

**Hussain Attia [1,*] and Fernando Delama [2]**

[1] Department of Electrical, Electronics and Communications Engineering, School of Engineering, American University of Ras Al Khaimah, Ras Al Khaimah P.O. Box 10021, United Arab Emirates

[2] Department of Sustainable Product Design and Architecture, Keene State College, Keene, NH 03435, USA; fernando.delama@keene.edu

* Correspondence: hattia@aurak.ac.ae

**Abstract:** There are many remote buildings that cannot be supplied by alternating electricity of the utility grid. Due to this, this study proposes adopting Direct Current (DC) appliances for a stand-alone remote building. Direct Current can be supplied from a suitable photovoltaic array which can harvest renewable solar energy. This proposal guarantees an efficient power system by removing the necessity of including an inverter, power filter, insulation transformer, and a complicated controller, which are usually needed for producing Alternating Current (AC) power to feed AC loads using a PV system. When the proposal is applied, the PV system will be more efficient, simple, affordable, and more compact. A detailed power requirement calculation for a typical house uses DC appliances, generalized steps to design a suitable PV array, and an Artificial Neural Network (ANN) algorithm for guaranteeing Maximum Power Point Tracking (MPPT); all of which are introduced for remote buildings. The main contribution of this paper is proposing an integrated design of a DC unit of 11 kW·h PV system for stand-alone buildings that eliminates three stages that improves the system performance compared to AC unit. The introduced study includes PV array calculation based on PV module of 220 W with an intelligent algorithm of four layers. The Mean Squared Error (MSE) of the proposed ANN equals $2.7107 \times 10^{-5}$ to guarantee a fast and accurate MPP tracking for continuously harvesting maximum power from the incident sunlight. An energy storage unit of 12 batteries 12 V/150 Ah of matrix dimensions $3 \times 4$ is designed in the DC unit for energy saving to feed the DC appliances during night hours. MATLAB/Simulink Version R2015b is used to simulate the introduced DC power unit and collect the testing records for analyzing the unit performance.

**Keywords:** stand-alone system; direct current appliances; PV array; ANN; MPPT; buck converter; MATLAB Version R2015b

## 1. Introduction

There are no doubts that renewable and clean energy sources have been introduced decades ago as an effective alternative to reduce dependence on traditional energy sources in order to reduce environmental pollution, global warming, and dependence on fossil fuels. Renewable energy sources have many advantages; the energy is available in all parts of our planet. It is infinite clean energy, in addition to being free of cost. Also, it does not need a transmission network from the site of generation to the site of consumption [1–3]. Solar energy represents the most important source of renewable energy, as solar energy is easily converted into direct current electricity through the photovoltaic (PV) panels. A PV panel contains a number of solar cells, which convert the energy of the incident light on their surface into proportional electricity. The produced electrical power is directly proportional to the amount of falling light, whereas it is inversely proportional to the ambient temperature. The levels of produced voltage and current are depending on the

number of solar cells involved in the PV module and the way of connection, whether in series or parallel. The total power delivered from a panel equals to the summation of the delivered power from each solar cell of the PV panel [4,5].

The vast availability of solar energy adds another merit related to the distant buildings from the utility grid; this merit is represented by the possibility of delivering the required electricity using stand-alone PV system to these buildings. This effective alternative is able to supply the required electricity without the need for an additional infrastructure, such as expanding the traditional grid to the far building [6,7].

To design a stand-alone PV system, the limitation of solar energy availability outside day hours has to be taken into consideration. The electricity is required for 24 h per day for electrical appliances of a building or house. An energy storage unit such as a battery bank is important to be involved in the system to store sufficient electrical energy to ensure delivering it during the night hours to the appliances [8,9]. Stand-alone PV systems that are used to supply AC appliances have important disadvantages, which include, firstly, the necessity of using an inverter of a suitable rated power, inserting a power filter, and step up insulation transformer. The second disadvantage is that the excessive stages of the PV system will increase the complexity and total cost, while simultaneously decreasing the overall system efficiency.

To optimize the level of harvesting energy from PV panels during the day hours, many researchers have studied the nonlinear behavior of PV panels in terms of load current and power, with respect to the output voltage. The researchers found that there is a certain working point at which maximum power can be harvested from the panel if the panel voltage is located in the reference voltage point. Based on that, the researchers focused on proposing different algorithms to guarantee accurate and fast tracking to the Maximum Power Point (MPP). Regardless of the type of PV system, whether it is a stand-alone or grid-connected system, guaranteeing MPP tracking condition maximizes the level of harvested energy from the incident sunlight [10–28]. The proposed algorithms have different characteristics, including both pros and cons, in terms of response speed, tracking accuracy, and oscillation at steady state conditions. Perturb and Observe (P&O) algorithm and Incremental Conductance (IC) algorithm are the common traditional algorithms [10] used for MPP tracking. Steady state oscillations and a slow response in MPP tracking are the two main disadvantages of P&O and IC algorithms, respectively.

Fuzzy Logic Control (FLC) is introduced for MPP tracking function to overcome the mentioned disadvantages of P&O and IC through a quick and stable response using fuzzification, Fuzzy rules, and defuzzification processes [11–14]. A new global MPP tracker combined FLC and an algorithm of scanning and storing process presented in [15] for stand-alone PV systems that work under partially shaded weather conditions.

Artificial Intelligence (AI) represented by ANN algorithm is a machine learning algorithm, which works by manipulating the data of input variables to have an accurate prediction for the missed values of the required variables. ANN is playing an effective role for the objective of guaranteeing MPPT in PV system applications by controlling different types of DC-DC converters [16–23].

Different structures of neuro-fuzzy in terms of memberships and parameter values are studied in [24] to evaluate the effectiveness of MPP tracking, and to reveal the merits and demerits of these structures. In the study [25], two new ANN based MPP tackers are introduced through fixed and variable step size, and the tracker performance is analyzed and evaluated. The hybrid processes also proposed in [26–28] merge ANN and FLC to guarantee the fast prediction and smooth system response during parameter variation to improve the MPP tracking function.

Based on the above-mentioned, this paper focuses on improving the performance of the PV systems that feed the remote stand-alone buildings, through designing an effective DC power delivering PV unit. This study demonstrates detailed design steps, including power calculation of a typical house, design stages of PV array based on PV module of 220 W, design a $3 \times 4$ matrix of battery bank using 12 V/150 Ah batteries for energy saving,

proposing a suitable ANN algorithm for quick and effective MPP tracking function. The contribution of this paper proposes an effective solution for a remote building based on DC appliances and an intelligent MPP tracker. This paper offers an integrated solution that can supply the required electrical power to DC appliances of a typical house or building located in a remote area from the utility grid. Firstly, the suitable type and quantity of DC appliances to be used for a typical house are proposed, as well as a detailed power calculation and design steps of a certain power-delivering unit of 11 kW·h, through an artificial intelligence MPPT. The remainder of this paper is as follows: Section 2 shows the advantages of adopting DC appliances rather than AC. Detailed power calculation of a sample of DC loads that can be involved in a typical building are shown in Section 3. Section 4 demonstrates the block diagram of the proposed solution. Generalized steps of designing a PV array are shown in Section 5, while Section 6 explains the design of a DC-DC buck converter that will be used in the presented PV unit. Section 7 shows the details of the designed ANN algorithm for MPPT purpose. Simulation results with their analysis and discussing the merits of the proposed system are explained in Section 8. Finally, the summary of conclusion and future points for improvement are discussed in Section 9.

## 2. Advantages of Adopting a PV System for DC Instead of AC Loads

This study proposes a DC unit-based PV system for delivering power to stand-alone buildings. Compared to an AC unit, the proposed DC unit noticeably reduced the PV system stages. Figure 1 shows the stages of a stand-alone PV system supplying AC Load, whereas Figure 2 shows the stages of a stand-alone PV system supplying DC Load. The difference between AC and DC units in terms of involved stages has clearly compared in Figures 1 and 2. To deliver the required electricity from solar PV panels to an AC load, the first stage of the system is a PV array. This array is harvesting the energy from the incident sunlight by converting the solar energy to electrical energy positively proportional with the light intensity and a negatively proportional with ambient temperature. The second stage is a DC-DC converter with a storage unit used to convert the fluctuated output voltage of the PV array to regulated voltage and store the harvested energy in a battery bank. The third stage is an MPPT unit, which is involved in the system to maximize the harvested energy by tracking the reference voltage. At the reference voltage, the maximum power (MP) point is located and can be tracked. Based on the instantaneous value of the reference voltage, the duty ratio of a Pulse Width Modulation (PWM) drive pulses will be controlled. DC to AC inverter represents the fourth stage of the system, the function of the inverter is to invert the DC electrical power to AC to allow it to be suitable for AC loads. Low Pass Filter (LPF) of power components is necessary to enhance the shape of output AC voltage in terms of Total Harmonic Distortion (THD) and harmonics from the inverter voltage and current. The function of the last stage is to step up the AC voltage to a suitable level to the connected AC load. In comparison to the PV system supplying AC load, the PV system supplying DC load includes lower stages, and is capable of dispensing three stages, namely, DC-AC inverter, power filter, and insulation transformer. Due to this elimination, the DC system guarantees higher efficiency [29], lower system complexity, lower implementation time, lower probability of maintenance, longer working period, smaller system size, and lower cost.



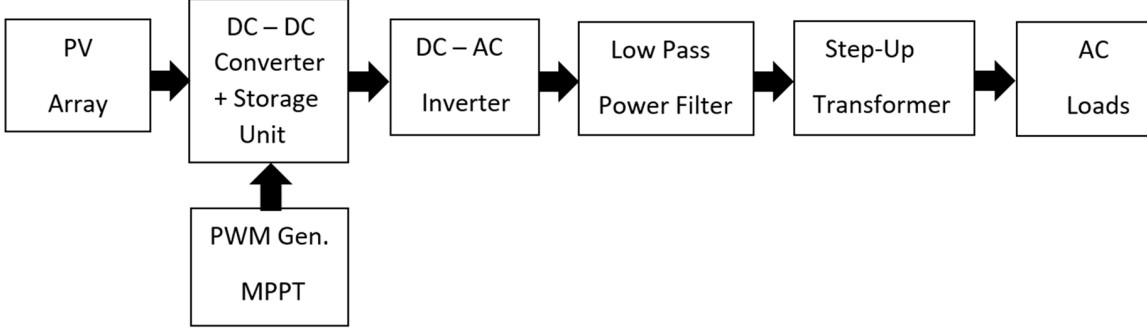

**Figure 1.** Block diagram of stand-alone PV system supplying AC loads.

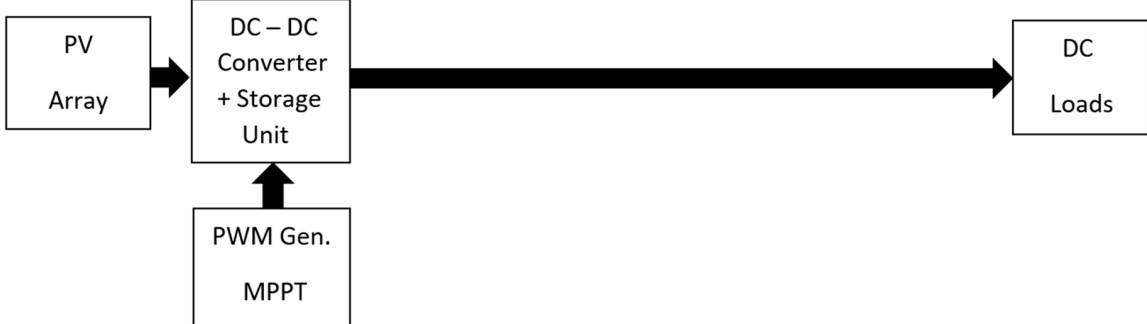

**Figure 2.** Block diagram of stand-alone PV system supplying DC loads.

### 3. Power Calculation

The necessary electrical DC appliances are considered in calculating the total required power from the proposed PV unit. Normally, the appliances include stove, air conditioner, refrigerator, television, lighting lamps, DC fans, and DC water pump. For energy saving, it is important to select low power consuming appliances. For example, the type of lighting units should be Light Emitting Diode (LED), and the TV should be a smart television with Liquid Crystal Display (LCD) or LED display.

Table 1 demonstrates the types of electrical appliances, required quantity, rated power, total working hours per day of each appliance, and the total required daily power. Based on the load details, the total required power per day is 10.5 kW·h. The storage unit (battery bank) capacity is 10.8 kW·h, which should be greater than the required power, and the harvested energy from the system PV array should be greater than the storage unit capacity. Based on that, the proposed PV array in this study is 11 kW·h, which is designed using a PV module of MPP power 220 W.

**Table 1.** Calculations of required power per day.

| Equip. and Device | Storage Unit, Battery Bank | Load 1: Stove | Load 2: Air Conditioner | Load 3: Fridge | Load 4: TV | Load 5: LED Light | Load 6: DC Fan | Load 7: DC Pump |
|---|---|---|---|---|---|---|---|---|
| Qty | Batteries matrix 3 × 4 × 12 V/150 A·h | 1 | 2 | 1 | 1 | 5 | 2 | 1 |
| Rated Power (W) | Normal Battery Charging = 10% of Battery Capacity [30], 10% × 150 A = 15 A × 3 Rows = 45 A | 1000 | 1000 | 500 | 200 | 10 | 50 | 200 |
| Duty Hours/day | Charging 5 h/day | 1.5 | 2 | 6 | 5 | 10 | 2 | 1.5 |
| W·h per day | 3 × 48 V × 15 A × 5 H = 10,800 W·h Total storage community = 10,800 W·h | 1500 | 4000 | 3000 | 1000 Total required W·h per day = 10,500 W·h | 500 | 200 | 300 |

## 4. Block Diagram of Proposed System

The presented PV unit is able to provide electricity for stand-alone home appliances of total rated power up to 11 kW·h. The proposed unit will guarantee MPP tracking by an intelligent process using ANN algorithm. Figure 3 shows the block diagram of the introduced PV system. The system involves an array of 10 PV panels to produce a suitable level of DC-link voltage. This voltage will be delivered to the DC-DC buck converter, which reduces the level of the input voltage to a voltage level suitable for home appliances. Pulse width modulated pulses of 10 kHz drive the converter switch. An intelligent algorithm controls the duty ratios of these pulses to guarantee MPPT work conditions. A suitable matrix of $3 \times 4$ 12 V battery bank of current capacity 150 Ah is inserted in the presented DC unit to satisfy the required energy saving through normal charging [30]. The details of sub-functions will be explained in the next sections, whereas the details of designing the battery charger is out of the scope of this study.

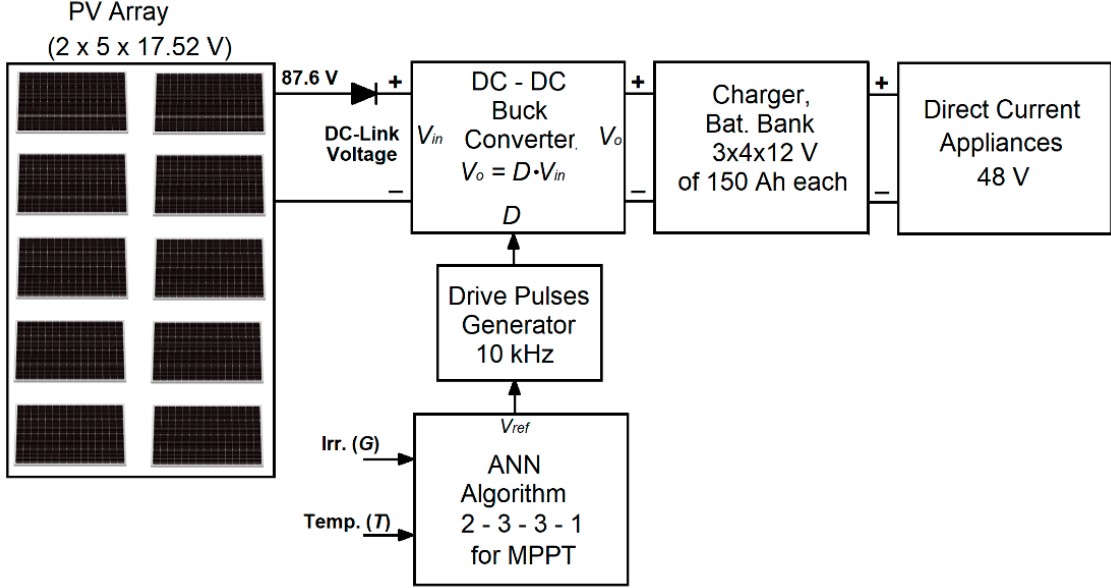

**Figure 3.** Block diagram of the proposed PV unit.

## 5. Design Procedure of Photovoltaic Array

*5.1. Characteristics of Selected PV Module*

The function of a solar cell is to produce an equivalent electrical power proportional with the level of incident sunlight, whereas it is negatively proportional with the ambient temperature. The relationships (1) to (7) below reflect the effects of delivered electrical current and voltage by the solar irradiation ($G$) and ambient temperature ($T$) to the solar cell. Figure 4 shows the equivalent electrical circuit of a solar cell. The photovoltaic PV panel is constructed by connecting a number of solar cells in a certain arrangement. To increase the output-generated voltage from the PV panel, solar cells should connect in series. To increase the delivered load current, a number of PV panels or PV strings should connect in parallel [28]. The selected PV module for the proposed PV array is Newpowa 220 W Monocrystalline 10BB Cell Solar Panel 220 Watt of a high efficiency from NewPowa, 3633 Inland, USA [31]. Table 2 shows the electrical specifications and the cell arrangement of the selected PV panel.

The proposed PV array should be able to deliver the required energy or more per day. Based on power calculation of the unit's load and the total required power, the PV array should be able to produce 11 kW·h as daily rated energy. Considering five sunny hours per day, the maximum power that should be generated from the PV array in this study is 11 kW·h/5 h = 2.2 kW.

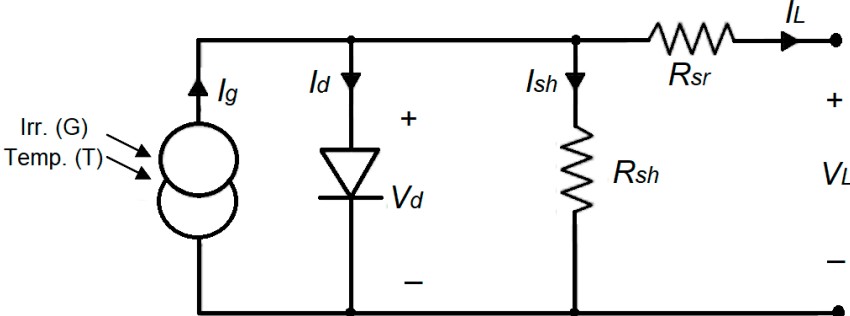

**Figure 4.** Electrical circuit of a solar cell.

**Table 2.** Electrical specification of Newpowa 220 W Monocrystalline 10BB cell solar panel.

| Parameter | Value |
|---|---|
| Max. Power Output (W) | 220 W |
| Voltage at MPP $V_{MPP}$ (V) | 17.52 V |
| Current at MPP $I_{MPP}$ (A) | 12.6 A |
| Voltage Open Circuit $V_{OC}$ (V) | 20.52 V |
| Current Short Circuit $I_{SC}$ (A) | 13.41 A |
| Cells Arrangement (X, Y) | $15 \times 4$ Solar cells |

Equations (1)–(7) demonstrate the relationships of generated current ($I_g$) inside solar cell, the load current ($I_L$) which is delivered from the output terminals of the cell, diode current ($I_d$), shunt resistor ($R_{sh}$) current ($I_{sh}$), and output load voltage ($V_L$), which are calculated after considering the drop voltage of a serial resistor ($R_{sr}$). In addition to the relationships of how to calculate total delivered voltage ($V_{DC\text{-}Link}$) based on the number of serially connected cells $N_{serial}$, and how to calculate the total load current ($I_{L\_sys}$) when a number of serial lines are connected in parallel $N_{parallel}$:

$$I_L = I_g - I_d - I_{sh} \tag{1}$$

$$I_g = \frac{G}{G_{ref}}\left(I_{g\_ref} + K_{SCT}\left(T_c - T_{c\_ref}\right)\right) \tag{2}$$

$$I_d = I_o\left[e^{\frac{V_d}{V_t}} - 1\right] \tag{3}$$

$$I_{sh} = \frac{V_d}{R_{sh}} \tag{4}$$

$$V_L = V_d - R_{sr}I_L \tag{5}$$

$$V_{DC\_Link} = N_{serial} \times V_L \tag{6}$$

$$I_{L\_Sys} = N_{parallel} \times I_L \tag{7}$$

The behavior of the selected PV module (Newpowa 220 W Monocrystalline 10BB) is simulated using MATLAB/Simulink Version R2015b. The simulated curves of load current and load power with respect to the output voltage of the PV module are plotted in Figures 5–8. Figures 5 and 6 demonstrate the current and power curves for different levels of sunlight intensity (200 W/m², 400 W/m², 600 W/m², 800 W/m², 1000 W/m²) and fixed ambient temperature (25 °C). Figures 7 and 8 demonstrate the current and power curves for different levels of ambient temperature (5 °C, 15 °C, 25 °C, 35 °C, 45 °C,) and fixed sunlight intensity (1000 W/m²).

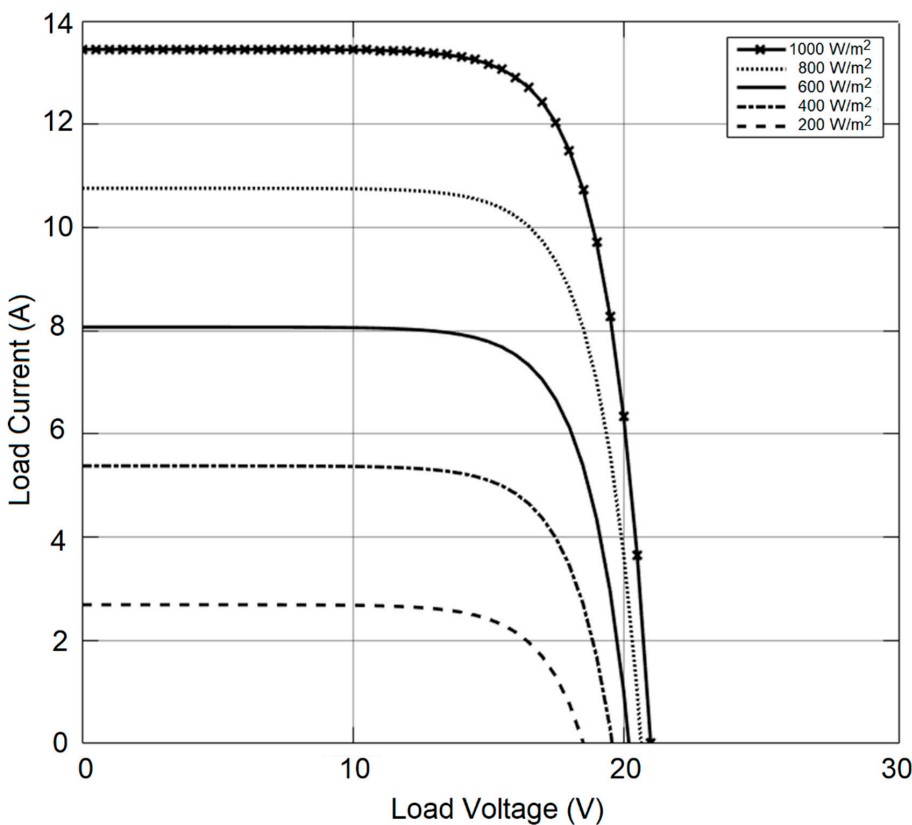

**Figure 5.** PV module behavior of output load current with respect to output load voltage at different sunlight irradiation and fixed ambient temperature (25 °C).

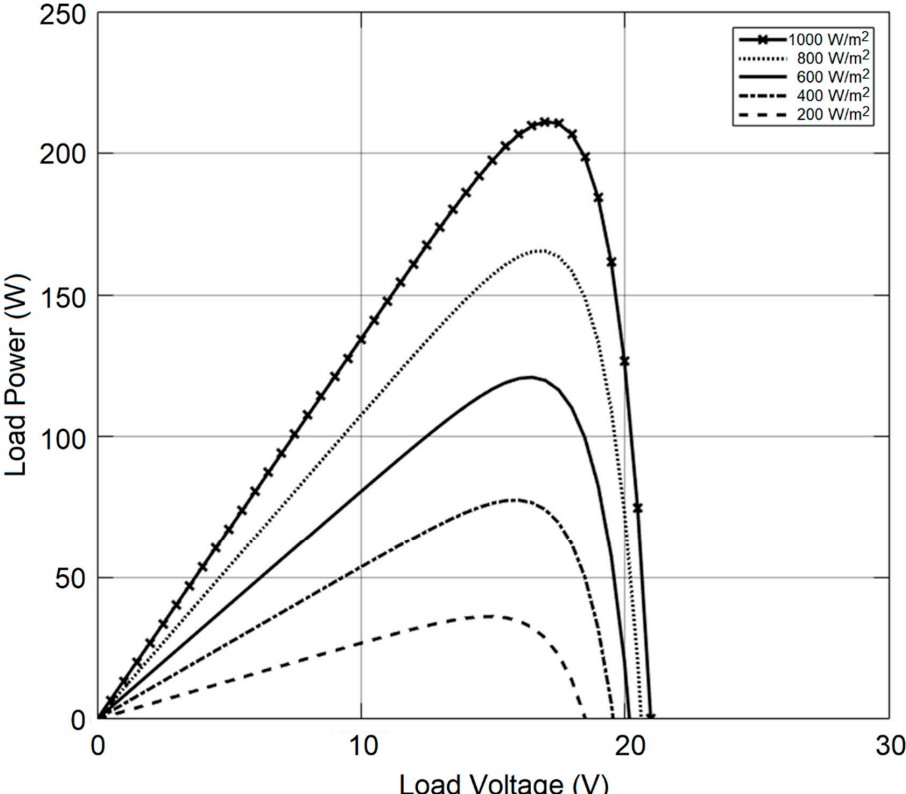

**Figure 6.** PV module behavior of output load power with respect to output load voltage at different sunlight irradiation and fixed ambient temperature (25 °C).

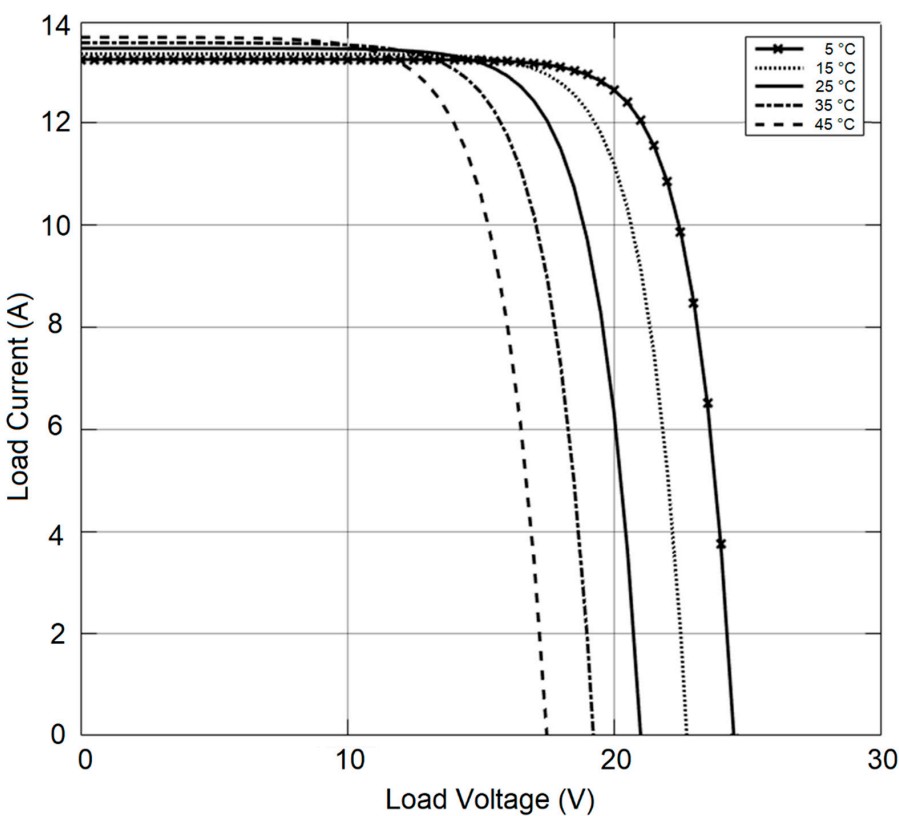

**Figure 7.** PV module behavior of output load current with respect to output load voltage at different ambient temperature and fixed sunlight irradiation (1000 W/m$^2$).

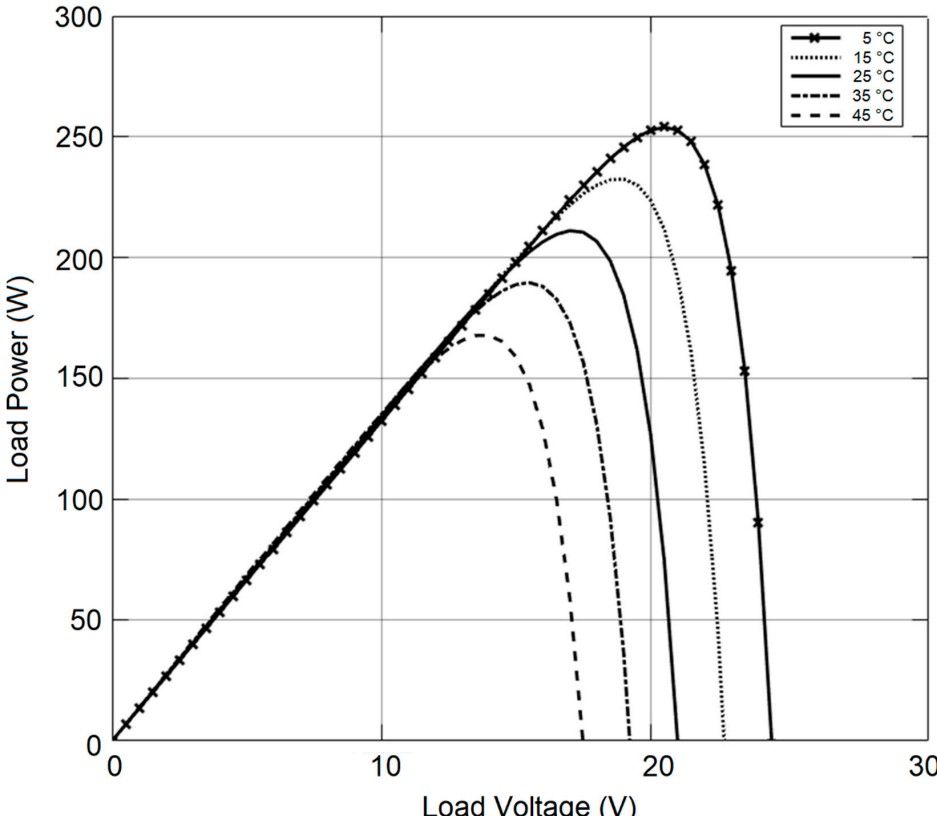

**Figure 8.** PV module behavior of output load power with respect to output load voltage at different ambient temperature and fixed sunlight irradiation (1000 W/m$^2$).

### 5.2. Design of PV Array

To guarantee delivering 11 kW per five hours daily or 2.2 kW per hour. The PV array should be able to deliver 2.2 kW at MPPT working conditions. To satisfy this request, the specifications of the selected PV module should be taken into consideration. Normally, big size PV array of Kilowatts range includes many PV strings, whereas the number of PV panels that are serially connected in one string is depending on the output voltage from one panel ($V_{mpp}$) at MPPT conditions and depending on the total required DC-link voltage:

$$PV_{Modules} \text{ Serially connected in a string } (N_{PV\_Serial}) = \frac{V_{DC\_Link}}{V_{mpp}} \tag{8}$$

Based on the result of (8), the actual number of serially connected PV panels ($N_{int\_Serial}$) should be an integer number greater than or equals to $N_{PV\_Serial}$:

$$N_{int\_Serial} \geq N_{PV\_Serial} \tag{9}$$

So, the actual generated DC-Link voltage ($V_{.ActDC\_Link}$) will be determined based on the actual number of the serially connected PV panels ($N_{int\_Serial}$):

$$V_{ActDC\_Link} = N_{int\_Serial} \times V_{mpp} \tag{10}$$

The total power that can be delivered by one PV string ($P_{PV\_String}$) equals to the power delivered by one panel at MPP conditions ($P_{mpp}$) multiplied by the actual integer number of serially connected panels in the PV string ($N_{int\_Serial}$):

$$P_{PV\_String} = P_{mpp} \times N_{int\_Serial} \tag{11}$$

To produce the total rated power from the PV system ($P_{Sys}$), the number of strings ($N_{PV\_Strings}$) that are in parallel should equal the total rated power divided by the power delivered by one string, or:

$$N_{PV\_Strings} = \frac{P_{Sys}}{P_{PV\_String}} \tag{12}$$

The actual number of PV strings ($N_{Act\_PV Strings}$) should be an integer number greater than or equal to the result of (12):

$$N_{int\_PV Strings} \geq N_{PV\_Strings} \tag{13}$$

The total power that can actually be delivered by the PV system ($P_{PV\_Sys}$) equals to the power delivered by one PV string multiplied by the actual integer number of the connected PV strings in the system:

$$P_{PV\_Sys} = P_{PV\_String} \times N_{int\_Strings} \tag{14}$$

Figure 9 shows generalized flowchart steps for designing a PV array of any DC-link voltage and MPPT power delivering range. The flowchart starts by entering the values of PV module voltage and power at the MPP conditions, and the required DC-link voltage and rated power of the system. Calculating and approximating the number of PV modules that are serially connected in one string are processed in the second and third steps of the flowchart, respectively. The calculation of the rated power that can deliver by one string is processed in the fourth step. The number of PV strings that are connected in parallel are calculated through the last two steps of the flowchart.

Figure 10 shows the way of connecting the proposed PV array, the array involves two strings, each one includes five PV module of 220 W. The proposed connection involves bypass diodes to avoid the damaged modules and let the remaining serial modules continue working. Blocking diodes are also included in the connection to block any damaged PV string. Based on the module specifications, the DC-link voltage that can be found on the

two terminals of the array equals to 87.6 V at the MPP condition. The total rated power that can be harvested by the array equals to 2.2 kW.

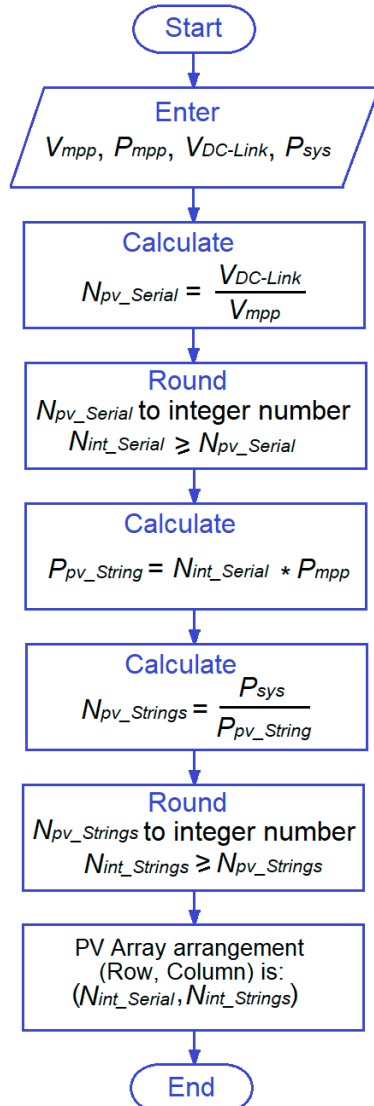

**Figure 9.** Generalized flowchart for designing a PV array.

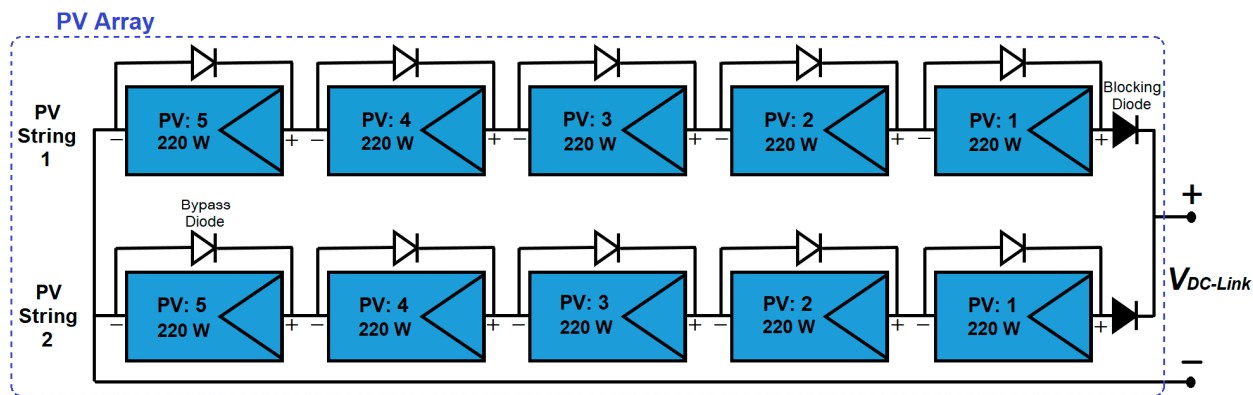

**Figure 10.** Proposed serial-parallel arrangement of the PV array to deliver 11 kW daily.

## 6. Design of DC-DC Buck Converter

The function of a DC-DC converter, after receiving the input DC-link voltage, is to control and stabilize the fluctuated input voltage to a regulated voltage level. There are

different types of converters that can be used to step-up or step-down the fluctuated or unregulated voltage input to a certain regulated level. The fundamental types of converters, namely, are the buck converter to step down the input voltage, the boost converter that is used to step up the input voltage, and the buck-boost converter which is able to step down or step up the input voltage depending on the value of duty cycle [32].

This study uses a DC-DC buck converter in the design of the DC Unit; Figure 11 shows the arrangement of the electrical components of the selected converter. Buck converter uses passive components such as inductor, capacitor, and resistor, whereas it uses semiconductor components such as diode and transistor. PWM drive pulses generator is used to drive the converter's switch. The duty ratio $D$ of these pulses is controlled to regulate the output voltage of the converter. The converter serves as a DC transformer.

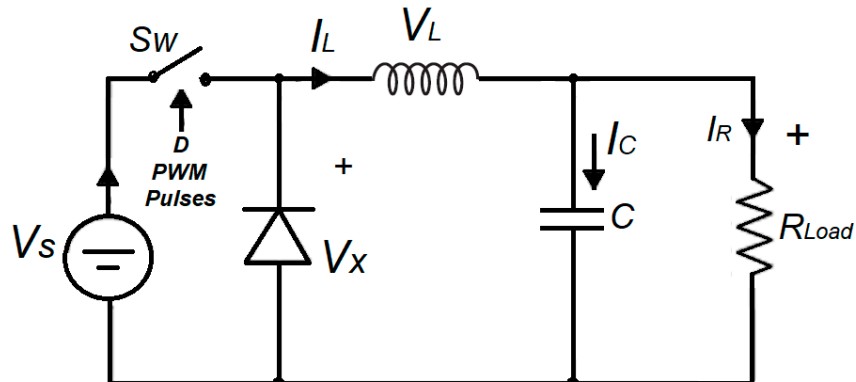

**Figure 11.** The circuit arrangement of DC-DC Buck converter.

The operation analysis of buck converter will consider the steady state of the circuit; therefore, it will be working in a Continuous Current Mode (CCM). In other words, the inductor current will be continuous and always in a positive direction. The inserted capacitor in the converter is large enough to have a constant load voltage. In addition, all the circuit components used in the presented system are ideal in terms of function and specifications.

The selected switching frequency ($f_s$) is a second face to the switching period ($T_s = 1/f_s$). Switching period ($T_s$) is the summation of the closing and opening time of the converter switch [$T_s = DT_s + (1 - D) T_s$].

The analysis starts with considering the net change of the inductor's current, and the average inductor voltage equals to zero after each full switching period. Figure 12 shows the equivalent circuit of the converter when the switch is closed in a duration $DT_s$.

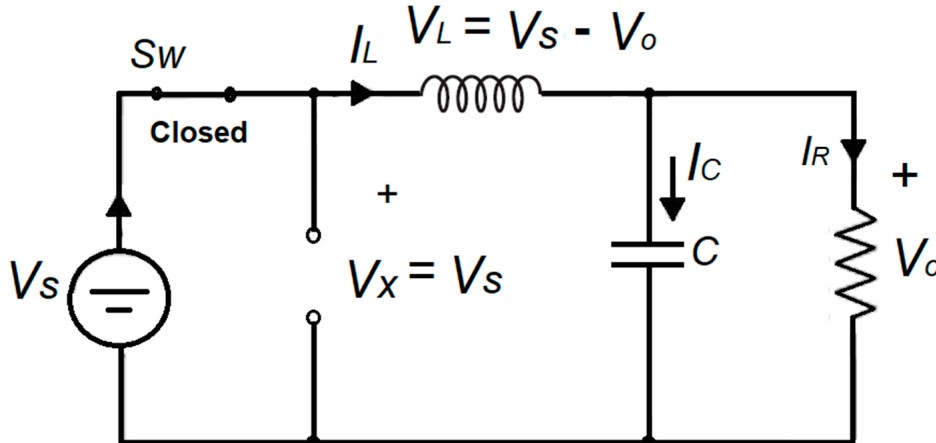

**Figure 12.** The equivalent converter circuit during switch closing.

While the switch is closed, the voltage across the inductor equals the difference between the supply voltage and the output voltage, because the diode is open and it is forced to be reversed by the effect of the DC source [32]:

$$V_L = V_s - V_o = L\frac{dI_L}{dt} \tag{15}$$

Rearranging (15) and considering the duration of switch's closing time $DT_s$:

$$(\Delta I_L)_{closed} = \left(\frac{V_s - V_o}{L}\right)DT_s \tag{16}$$

Figure 13 shows the equivalent circuit of the converter when the switch is opened of duration $(1 - D)T_s$.

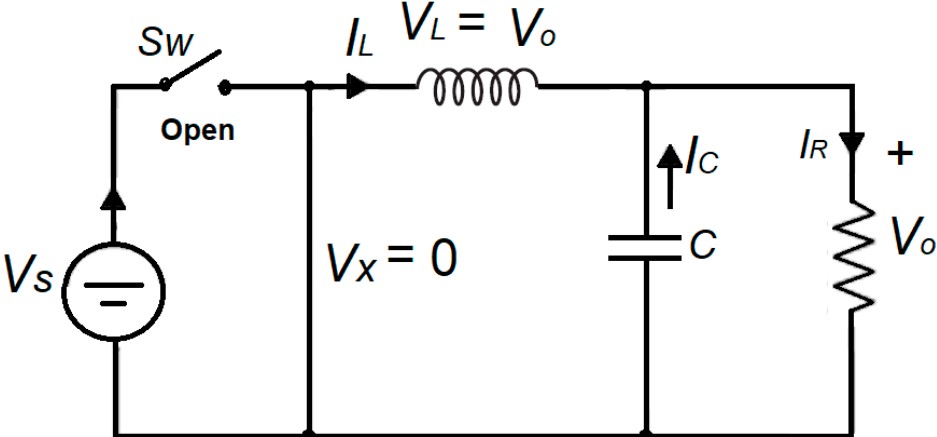

**Figure 13.** The equivalent converter circuit during switch opening.

During the opening time of the switch, the inductor voltage will be the opposite polarity of the output load voltage because the diode is forward-biased, so:

$$V_L = -V_o = L\frac{dI_L}{dt} \tag{17}$$

Rearranging (17) and considering the duration of switch opening time $(1 - D)T_s$:

$$(\Delta I_L)_{open} = -\left(\frac{V_o}{L}\right)(1 - D)T_s \tag{18}$$

After the full switching period, the inductor current will be at the same value as before starting the period, so the summation of the inductor current variations will be zero, or:

$$(\Delta I_L)_{closed} + (\Delta I_L)_{open} = 0 \tag{19}$$

By replacing the inductor current variation, yields

$$\left(\frac{V_s - V_o}{L}\right)DT_s - \left(\frac{V_o}{L}\right)(1 - D)T_s = 0 \tag{20}$$

or

$$V_o = V_s D \tag{21}$$

From (21), the value of the output voltage with be less than the value of the input voltage in a buck converter. The output voltage can be controlled by controlling the duty cycle $D$. For the steady state condition, the average current passing through the capacitor

will be zero, so the average inductor current is equal to the average load current or resistor current, so

$$I_L = I_R = \frac{V_o}{R} \tag{22}$$

Whereas the minimum inductor current is

$$I_{min} = I_L - \frac{\Delta I_L}{2} \tag{23}$$

Substituting the inductor current and the current variation yields:

$$I_{min} = V_o \left( \frac{1}{R} - \frac{1-D}{2Lf_s} \right) \tag{24}$$

For a converter working in CCM, the minimum inductor current should be more than zero, and so, to determine the minimum value of the converter inductor, equating Equation (24) to zero yields:

$$L_{min} = \frac{(1-D)R}{2f_s} \tag{25}$$

From (25), the selected value of the converter inductor should be greater than the minimum value.

To calculate the suitable value of the capacitor, the capacitor current must be positive and the variation in the output voltage equals to the variation of the capacitor charges ($Q_C$) divided by the capacitor value, or

$$\Delta V_o = \frac{\Delta Q_c}{C} \tag{26}$$

Based on the fact that the change of capacitor charges ($\Delta Q_C$) equals the area of the triangle of the charge variation with respect to the time [31], so

$$\Delta Q_c = \frac{T_s \, \Delta I_L}{8} \tag{27}$$

Combining (26) and (27) yields

$$\Delta V_o = \frac{T_s \, \Delta I_L}{8C} \tag{28}$$

Replacing the value of inductor current variation ($\Delta I_L$) in (28), and switching frequency instead of switching time yields the value of the required capacitor for the desired voltage ripple ($\Delta V_O/V_O$):

$$C = \frac{1-D}{8 \, L \left( \frac{\Delta V_o}{V_o} \right) f_s^2} \tag{29}$$

## 7. Neural Network Algorithm for MPPT Applications

The aim of using Neural Network algorithm in this study is to ensure fast and accurate tracking to the maximum power point to harvest the maximum energy from the incident sunlight [33,34]. The function of Neural Network is to predict an accurate value of reference voltage of the selected PV module to ensure MPPT working conditions of the proposed PV system. The prediction of the resultant value will be continually varying depending on the instantaneous values of the input variables, which are the sunlight intensity and ambient temperature. Processing by neural networks goes through at least three stages or layers. The layers namely are the input layer for receiving variables, the hidden inner layer for internal processing, and one outer layer for producing reference voltage. Each layer consists of one or more neurons. Figure 14 shows the proposed ANN algorithm layers with number of neurons in each layer. The presented algorithm includes four layers, which are

an input layer with two neurons, two hidden layers with three neurons each, and an output layer with one neuron.

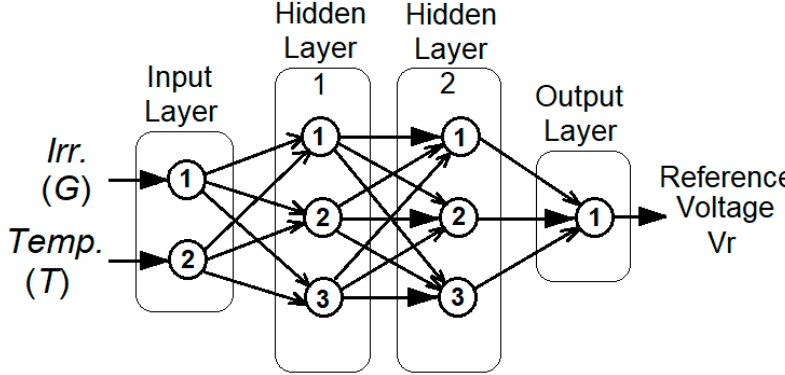

**Figure 14.** Proposed ANN algorithm for reference voltage predicting.

The structure of a neuron is shown in Figure 15, in which the neuron receives the input variables and multiply each one by an individual weight. All resultants will be added with a certain bias (*B*) in the neuron's adder. The output of the adder (*Z*) will be processed by an activation function $f(z)$ to generate the output result ($y_n$). This function can be one of three functions, namely, Linear bipolar, Sigmoidal or Hyperbolic function that are Equations (30), (31) and (32), respectively:

$$y_n = f(z) \tag{30}$$

$$f(z) = \frac{1}{1 + exp^{-z}} \tag{31}$$

$$f(z) = \frac{1 + e^{-2z}}{1 - e^{-2z}} \tag{32}$$

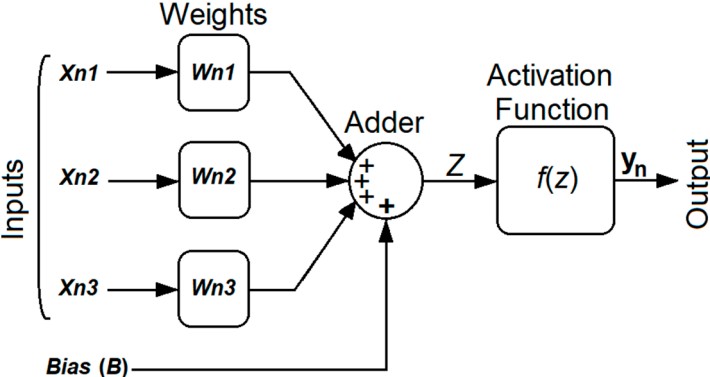

**Figure 15.** The structure of an Artificial Neuron.

The accuracy of ANN algorithm can be evaluated by monitoring the Mean Square Error (MSE) which indicates how the predicted values are far from the target ones. MSE represents the summation of error squares $e(k)^2$ of a certain number of input vectors (*Q*) divided by the vectors number.

$$MSE = \frac{1}{Q} \sum_{k=1}^{Q} e(k)^2 \tag{33}$$

The proposed ANN algorithm is characterized by the simplicity because it has a low number of hidden layers of low number of neurons to avoid the overfitting response. Overfitting prevents the perfect response because of the negative effect of environmental noise,

complex ANN layers and neurons, and/or limited training data. In other words, simple ANN structure will positively affect the algorithm performance in terms of overfitting.

## 8. Simulation Results Analysis and Discussion

System simulating considers the specifications of the selected PV module 220 W. MATLAB/Simulink Version R2015b is used to simulate the proposed PV unit. The first step of simulation is started by simulating the current and power curves with respect to the output voltage of the PV module. Many curves of load current and power are collected for different weather conditions. At different levels of sunlight intensity and fixed ambient temperature, Figures 5 and 6 show the current and power curves of the selected 220 W PV module. The curves demonstrate load current and power, which are positively proportional with the light intensity. Whereas Figures 7 and 8 show the current and power curves at different levels of ambient temperature and fixed sunlight intensity. The current curves start at approximately the same levels but decline at different voltages. This reflects on the power curves, which are started by the same slope with different levels of power then decline with different voltages, similar to current curves shown in Figure 7.

The designed PV array is simulate using MATLAB/Simulink Version R2015 for the system by considering the module specifications. To deliver a suitable DC-link voltage to the input side of the buck converter, the PV array is proposed to consist of five modules in serial connections to have PV string then two strings are connected in parallel. The designed ANN of four layers (input layer, two hidden layers, and output layer) is simulated in Figure 16a–c), which shows the details of the designed Feed-Forward Neural Network algorithm. The simulation of the ANN demonstrates the best validation performance of low MSE. MSE equals $2.7107 \times 10^{-5}$ at epoch 255 of 261 test epochs as shown in Figure 17. Figure 18 shows the parameter values of ANN training and testing process, which demonstrate satisfying performance records.

Figure 19 shows the simulation of the presented PV unit, including the designed DC-DC buck converter and ANN using MATLAB/Simulink Version R2015b, of a program of 50 μs sampling time. The converter parameters are tabulated in Table 3, whereas Table 4 shows specifications of the used computer and the version of MATLAB software Version R2015b used for this study.

**Table 3.** Parameters of buck converter.

| Converter Parameter | Parameter Value |
| --- | --- |
| Switching frequency | 10 kHz |
| Inductor | 4 mH |
| Capacitor | 2200 μF |
| Full load | 5 Ω |
| Half load | 10 Ω |

**Table 4.** Specifications of applied computer and software for the simulation.

| Hardware/Software | Type/Version |
| --- | --- |
| Computer processor | Intel Core i7—2670QM |
| Clock speed | 2.2 GHz |
| RAM memory size | 8 GB |
| Solid State Drive SSD | 256 GB |
| MATLAB Software | R2015b |

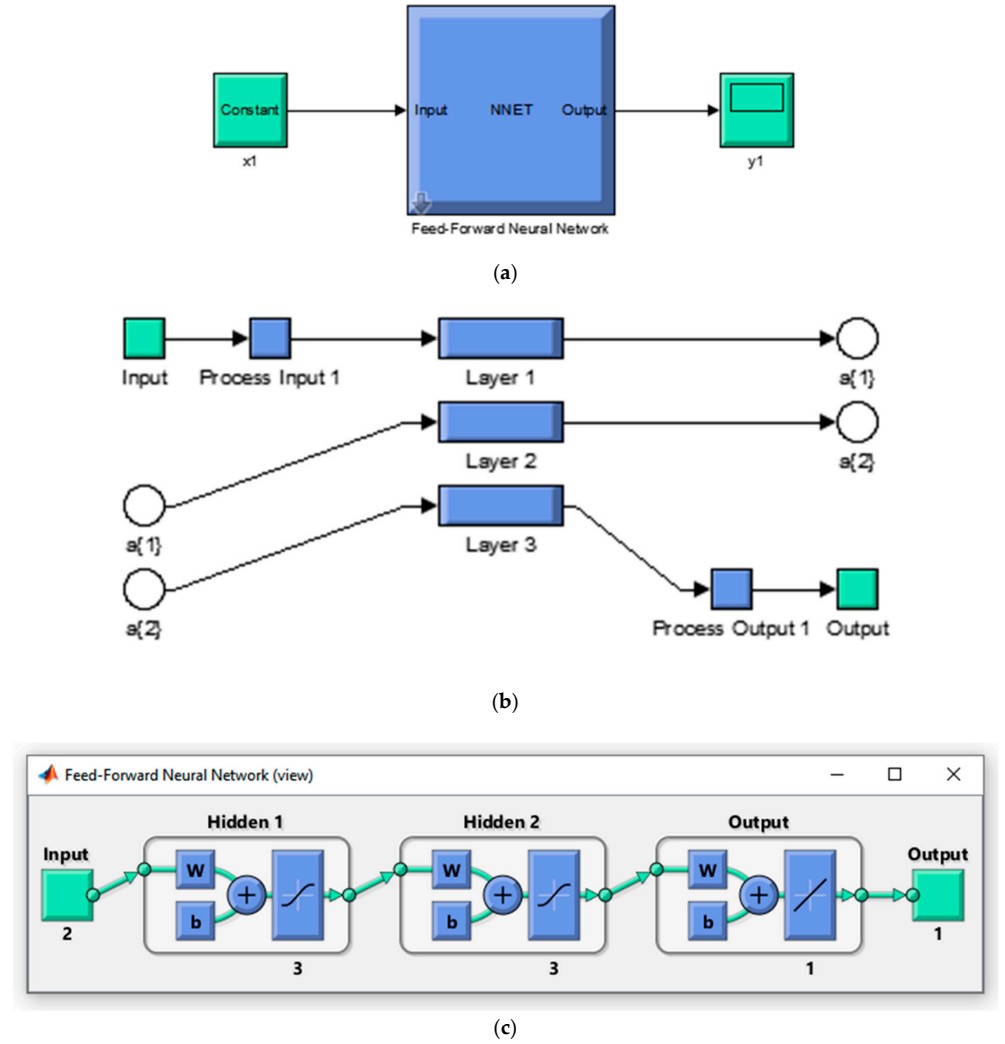

**Figure 16.** Simulated Feed-Forward Neural Network algorithm, (**a**) General structure, (**b**) Number of layers, (**c**) ANN layers contents.

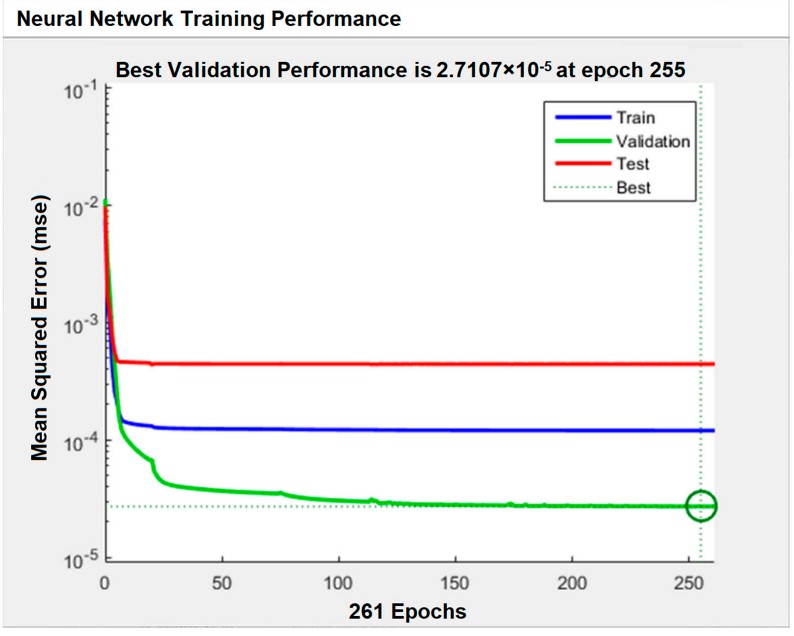

**Figure 17.** ANN training performance.

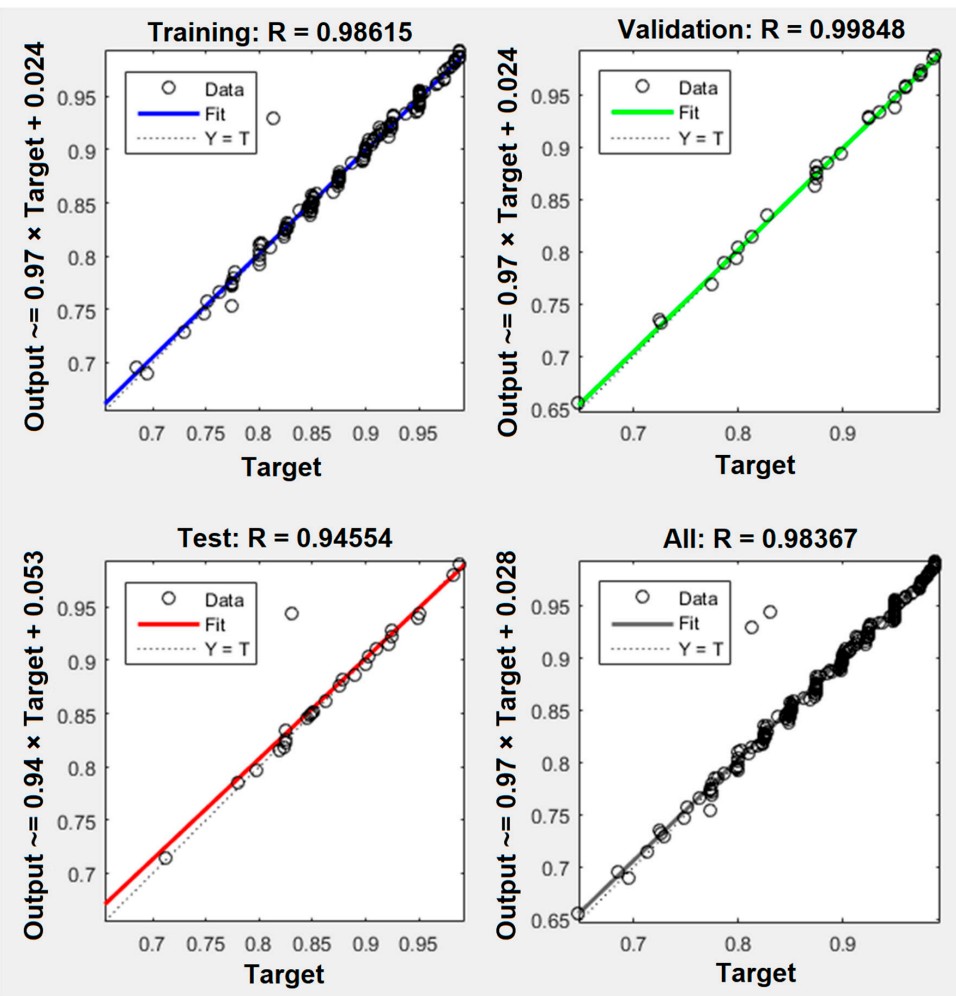

**Figure 18.** The training and testing process of the proposed ANN.

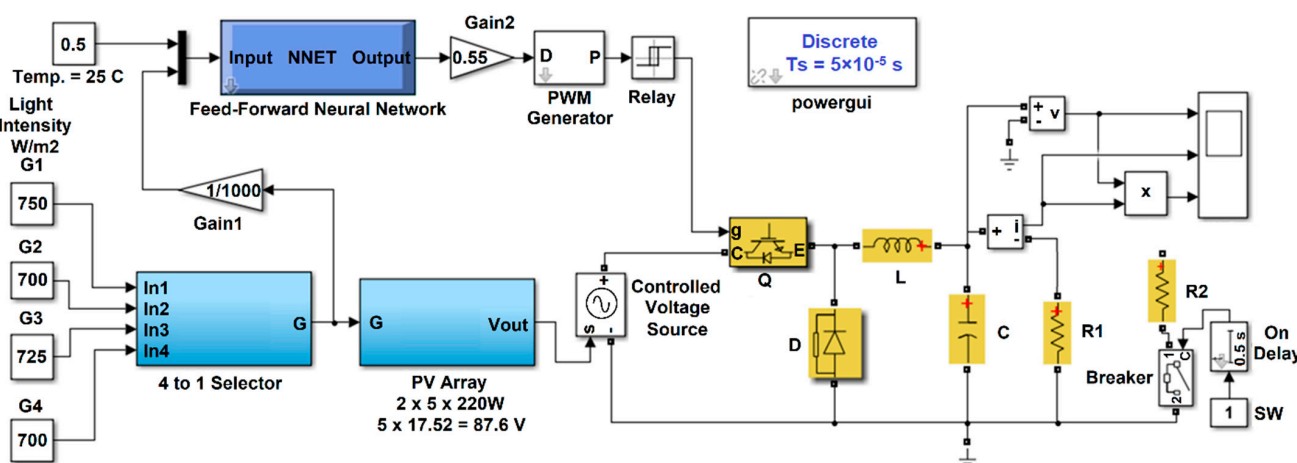

**Figure 19.** Simulation of the presented PV system.

The system simulation is run for 1 sec duration, which is divided into four equal divisions of 0.25 s each. Firstly, the system is tested with 50% of the rated load (10 Ω) with a range of sunlight intensities moving among 700 W/m², 725 W/m², and 750 W/m². After that, the test is repeated using full load (5 Ω). The system performance in terms of load voltage, load current, and load power are shown in Figures 20 and 21. Zooming in at 0.75 s confirms a steady load voltage with a very low voltage variation during light intensity variation.

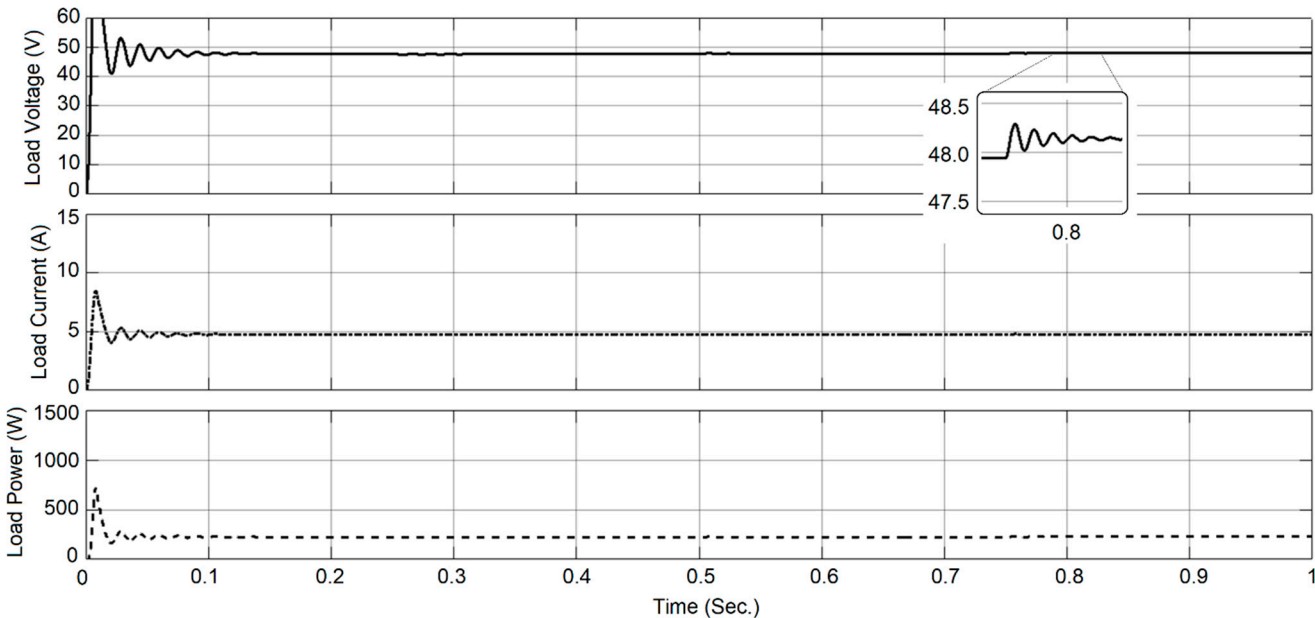

**Figure 20.** Simulation results of load voltage, current, and power at 50% load.

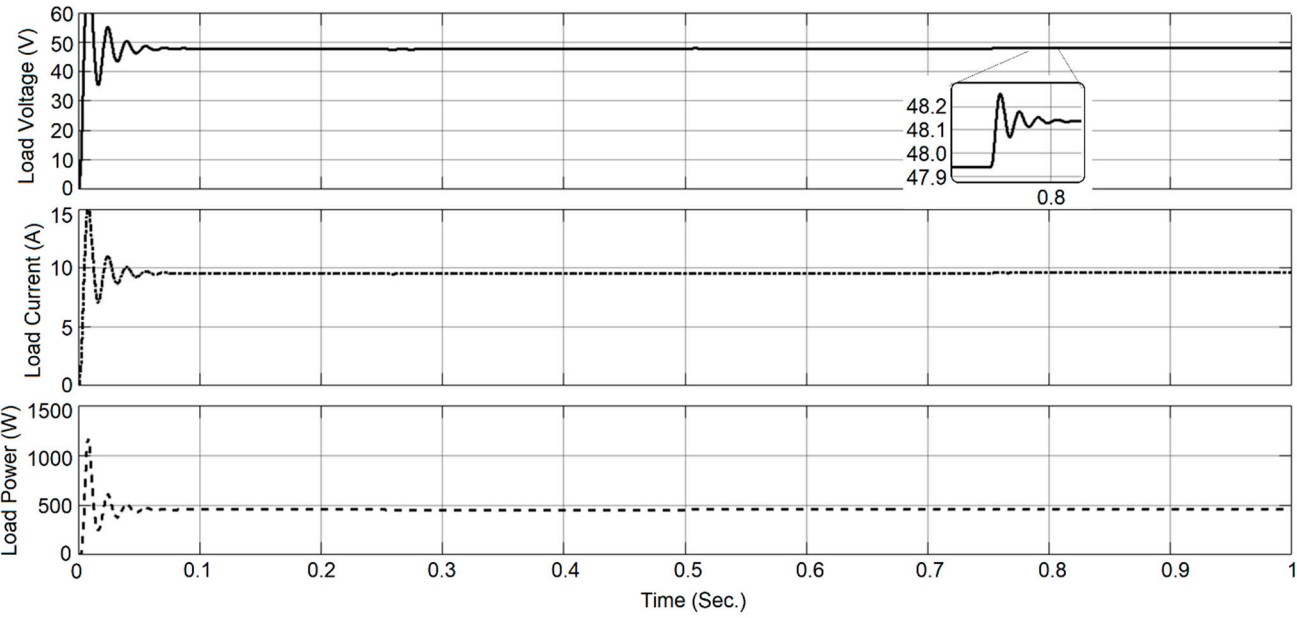

**Figure 21.** Simulation results of load voltage, current, and power at 100% load.

The performance of the system is also tested for the case of load variation from half to full load. The testing is performed for a simulation time of 1 second; in the first 0.5 s a half load is connected to the system; and then, the load is doubled to a full load for the next 0.5 s, as shown in Figure 22. The recovery time is measured at the instant of load variation and equals to approximately 40 ms.

In terms of discussing the merits of the presented solution, a multi-aspect comparison is addressed with the published research of [9,15,16,22,24,25]. In comparison, many sides are considered to have an accurate evaluation to this presented study. The comparison sides are: the system type whether it is AC or DC, stand-alone, or grid connected. System rated power, the calculation of the required power, stages involved in the system, size of the PV array, type of PV module, type of MPPT process or algorithm, layer number of ANN algorithm, and the value of MSE as an accuracy indication of ANN are all included in a comparison table. Table 5 shows a detailed comparison and demonstrates an integrated

presented work in terms of number of system stages, integrated calculation of power requirement that can be considered as a generalized calculation process of a multi- purpose PV system. On the other hand, a simple structure of ANN algorithm is proposed with a good MSE value to guarantee the maximum level of harvested power.

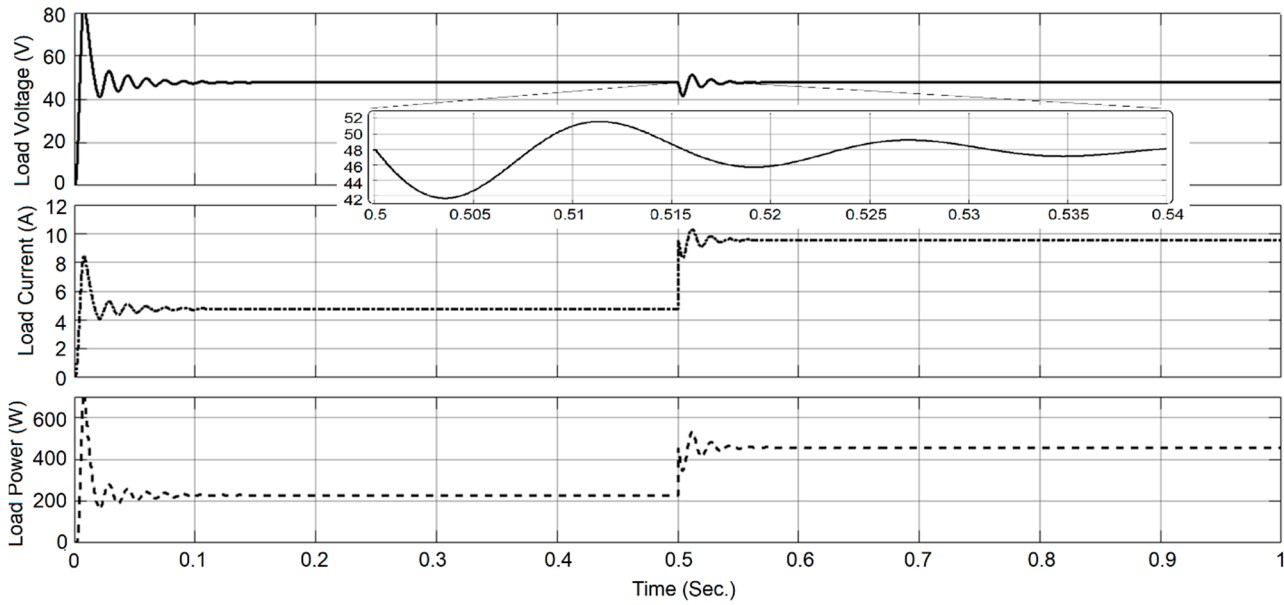

**Figure 22.** Simulation results during the load variation from 50% to 100%.

**Table 5.** Multi-aspect comparisons of the introduced solutions in the previous research.

| Ref. No., Year | [9], 2016 | [15], 2016 | [16], 2019 | [22], 2022 | [24], 2017 | [25], 2023 | Proposed PV System |
|---|---|---|---|---|---|---|---|
| System type | AC grid connected | DC Stand-alone | DC Stand-alone | AC Stand-alone | DC Stand-alone | DC Stand-alone | DC Stand-alone |
| Rated power | 480 W | 225 W, 600 W | 215 W | 12.5 kW·h | 240 W | 250 W | 11.0 kW·h |
| Loads power calculation | No | No | No | Yes | No | No | Yes |
| System stages | PV array, boost converter, MPP tracker, inverter, Step-up transformer, LC filter, load | PV array, boost converter, MPP tracker, load | PV array, boost converter, MPP tracker, load | PV array, MPP and charge controller, inverter, load | PV array, Flyback converter, MPP tracker, load | Solar panel simulator, MPP tracker, load | PV array, buck converter, MPP tracker, load |
| Size of PV array | $1 \times 2 \times 240$ W | $1 \times 3 \times 75$ W $1 \times 4 \times 150$ W | $1 \times 1 \times 215$ W | $9 \times 3 \times 345$ W | $1 \times 4 \times 60$ W | $1 \times 1 \times 250$ W Simulator | $2 \times 5 \times 220$ W |
| Type of PV panel | 240W-24V-Poly-Solar-Panel | LORENTZ 75W Shell SP 150-PC | ISTH-215-P | Mono-crystalline 345 W | Solarex MSX-60 | Kyocera KD 250 GX-LFB2 | Newpowa 220 W Mono-crystalline 10BB |
| MPPT Technique | P&O algorithm | FLC + Scanning and storing | ANN | Fast sweep algorithm | ANN | Neuro-Fuzzy | ANN |
| ANN layers | NA | NA | Three layers (2-5-1) | NA | Five layers (2-4-10-4-1) | Five layer (2-4-4-4-1) | Four layers (2-3-3-1) |
| ANN MSE | NA | NA | 0.39265 | NA | $9.4115 \times 10^{-6}$ | Many values of different algorithms | $2.7107 \times 10^{-5}$ |

## 9. Conclusions and Future Work

The full design steps of a DC power generation unit based on solar energy is proposed in this study for remote building applications. A photovoltaic unit of 11 kW is designed using a suitable arrangement of PV array and ANN algorithm. This study firstly demonstrated the advantages of adopting DC unit compared to AC unit, and how to avoid the need for an inverter, power filter, and insulation transformer, which are necessary in implementing an AC unit. This study presented a detailed power calculation of a DC unit including the desired electrical appliances with reasonable working hours. Generalized steps to design any size of PV array is presented in this study to have the desired DC-link

voltage and rated power. The specifications of the selected PV module, in terms of MPP voltage and power, are also considered in the design of the presented unit. An effective and fast response ANN algorithm of four layers (2-3-3-1) is designed for MPP tracking purpose to harvest maximum solar energy during day hours at any weather condition. The design steps of a DC-DC buck converter are shown to select converter parameters. MATLAB/Simulink Version R2015b software is used to simulate the system, collect the testing results, and evaluate the performance during one-second simulation duration of four equal divisions. The results confirmed the effectiveness of the presented unit in terms of load voltage and power at half and full load, and also in terms of recovery time which is approximately 40 msec at load variation. To improve the system performance in future research steps, a suitable controller, such as Proportional Integral Derivative (PID) or Sliding Mode Control (SMC), can be inserted in the system to smooth the shape of the load voltage signal and remove the spikes in load voltage and power. Secondly, involving a suitable battery bank to the system and replacing the Buck converter with a Buck-Boost converter to generalize the proposed DC unit application as a stand-alone system.

**Author Contributions:** Conceptualization, F.D.; methodology, H.A.; software, H.A.; validation, H.A.; formal analysis, H.A.; investigation, H.A.; resources, H.A.; data curation, H.A.; writing—original draft preparation, H.A.; writing—review and editing, H.A.; visualization, H.A.; supervision, H.A.; project administration, H.A.; funding acquisition, H.A. and F.D. All authors have read and agreed to the published version of the manuscript.

**Funding:** This research received no external funding.

**Institutional Review Board Statement:** Not applicable.

**Informed Consent Statement:** Not applicable.

**Data Availability Statement:** Not applicable.

**Acknowledgments:** The authors appreciate the financial support of the Office of Research and Community Service at the American University of Ras Al Khaimah, Ras Al Khaimah, United Arab Emirates, www.aurak.ac.ae.

**Conflicts of Interest:** The authors declare no conflict of interest.

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
