# Peer review of "Efficient Photovoltaic Unit for Power Delivering to Stand-Alone Direct Current Buildings Using Artificial Intelligence Approach Based MPP Tracker"

_sustainability, doi:10.3390/su151410861_

Round 1

Reviewer 1 Report

1. What are your contributions to the paper? (highlight them)

2. What are the merits compared to the existing approach? (prepare a comparison table) 

3. English language and style are fine/minor spell check is required.

4. Auther needs some more relevant literature reviews. prepare a summary table of it to improve the quality of the article.

Moderate editing of the English language required

Author Response

Manuscript ID sustainability-2472070.R1

“Efficient photovoltaic unit for power delivering to stand-alone direct current buildings using artificial intelligence approach based MPP tracker"

 Response to Reviewer 1

The authors would like to thank reviewer-1 for his/her constructive comments, which have undoubtedly improved the quality of the paper in its revision. In what follows, the reviewer’s remarks are in standard font, while the reply of the authors is in italics. Relevant changes in the paper are highlighted in GREEN.

Reviewer 1:

 Comments and Suggestions for Authors

 1. What are your contributions to the paper? (highlight them)

  • The paper’s contribution is mentioned in the abstract and other sections of the revised manuscript.
  1. What are the merits compared to the existing approach? (prepare a comparison table)
  • The merits of the existing approach are explained in the additional part of Section 8 and in Table 5 of the revised manuscript.
  1. English language and style are fine/minor spell check is required.
  • Editing of the English language is completed in the revised manuscript.
  1. Author needs some more relevant literature reviews. Prepare a summary table of it to improve the quality of the article.
  • Additional research studies’ outcomes are added to the introduction section, and related references are added to the reference list of the revised manuscript, as well as a comparison table of the presented work compared to other previous studies is included in the revised manuscript.

Comments on the Quality of English Language

 Moderate editing of the English language required

  • Editing of the English language is completed in the revised manuscript.

Reviewer 2 Report

This paper proposes the “Efficient photovoltaic unit for power delivering to stand-alone direct current buildings using artificial intelligence approach based MPP tracker”. The paper is well written but have few concerns and recommendations related to the manuscript: -

 1.      The author mostly used word “direct current DC”, used the bracket such as “Direct Current (DC)”. Once the abbreviation defines in the keywords or in paper, then it is not necessary to repeat it again, just write DC instead of writing direct current.

2.      It will be better to write “Direct Current” on Line ‘14’, instead of writing “Direct Current electricity”.

3.      Add some discussion related to the storage system connect with the PV system in the part of abstract.

4.      In the introduction part, the literatures cited are insufficient to support. Within this context, more studies (from 2022 and 2023) should be included and support your work with those studies.

5.      It is better to put the brackets when define some abbreviations, for detail see line ‘117’ total hormonic distortion THD, recommended (THD). Please do the changes as per recommendation for all other abbreviations.

6.      Figures 1 and 2, should be reconstructed because the word MPPT and some other words are missing in both figures. 

7.      In Table 1, in column of storage units which formulation the authors have used to calculate the battery current capacity. According to me, the calculations are ok but for the new reader it should be more elaborate or add some related equation in the paper. 

8. Figures 5-8, 15, 16 and 19-21 resolutions are not so clear. Put the figure in good resolution.

9. Equations 6-33 are not given in the proper format, should be in centered and remove the extra gaps.

10. The authors should concentrate the write paper (paragraph, table size, figures and captions) in proper formatting. For more details, please follow, the journal format. 

      1. The quality of English must be radically improved. There are many grammatical mistakes and typos in the submitted proposal.    

Author Response

Manuscript ID sustainability-2472070.R1

“Efficient photovoltaic unit for power delivering to stand-alone direct current buildings using artificial intelligence approach based MPP tracker"

Response to Reviewer 2

The authors would like to thank reviewer-2 for his/her constructive comments, which have undoubtedly improved the quality of the paper in its revision. In what follows, the reviewer’s remarks are in standard font, while the reply of the authors is in italics. Relevant changes in the paper are highlighted in GREEN.

Reviewer 2:

 Comments and Suggestions for Authors

 This paper proposes the “Efficient photovoltaic unit for power delivering to stand-alone direct current buildings using artificial intelligence approach based MPP tracker”. The paper is well written but have few concerns and recommendations related to the manuscript: -

  1. The author mostly used word “direct current DC”, used the bracket such as “Direct Current (DC)”. Once the abbreviation defines in the keywords or in paper, then it is not necessary to repeat it again, just write DC instead of writing direct current.
  • The authors thank the reviewer for this comment. The comment is followed in the revised manuscript.
  1. It will be better to write “Direct Current” on Line ‘14’, instead of writing “Direct Current electricity”.
  • The comment is followed on Line ‘14’ in the revised manuscript.
  1. Add some discussion related to the storage system connect with the PV system in the part of abstract.
  • Some discussions related to the storage system are added in the abstract and the content of the revised manuscript.
  1. In the introduction part, the literatures cited are insufficient to support. Within this context, more studies (from 2022 and 2023) should be included and support your work with those studies.
  • More studies from 2022 and 2023 are included in the introduction section of the revised manuscript, and added to the reference list [17]-[22], and [25].
  1. It is better to put the brackets when define some abbreviations, for detail see line ‘117’ total hormonic distortion THD, recommended (THD). Please do the changes as per recommendation for all other abbreviations.
  • The authors thank the reviewer for this comment. The comment is followed on Line ‘117’ and other parts of the revised manuscript.
  1. Figures 1 and 2, should be reconstructed because the word MPPT and some other words are missing in both figures.
  • Figures 1 and 2 are fully reconstructed in the revised manuscript.
  1. In Table 1, in column of storage units which formulation the authors have used to calculate the battery current capacity. According to me, the calculations are ok but for the new reader it should be more elaborate or add some related equation in the paper.
  • Normal Battery Charging = 10% of Battery Capacity [30]. The relation is included in Table 1, and Ref. [30] is added to the reference list of the revised manuscript.
  1. Figures 5-8, 15, 16 and 19-21 resolutions are not so clear. Put the figure in good resolution.
  • The authors thank the reviewer for the comment. The resolution of Figures 5-8, 15, 16 and 19-21 are increased to be clearer in the revised manuscript.
  1. Equations 6-33 are not given in the proper format, should be in centered and remove the extra gaps.
  • The equations are centered, and the extra gaps between the equations are removed in the revised manuscript.
  1. The authors should concentrate the write paper (paragraph, table size, figures and captions) in proper formatting. For more details, please follow, the journal format.
  • The authors thank the reviewer for the comment. The revised manuscript followed the comment.

 Comments on the Quality of English Language

 1. The quality of English must be radically improved. There are many grammatical mistakes and typos in the submitted proposal

  •  Editing of the English language is completed in the revised manuscript.

Reviewer 3 Report

Title: Efficient photovoltaic unit for power delivering to stand-alone direct current buildings using artificial intelligence approach based MPP tracker

Recommendation: Minor revisions needed as noted.

a

1. Is there a published article that is similar in the literature?

2. In the introduction section, it must be made apparent how the present work is innovative. The results of the investigation are not clearly explained.

3. What is the difference between your work and this paper (A new intelligent MPPT method for stand-alone photovoltaic systems operating under fast transient variations of shading patterns)?

4. Carefully check the superscript and subscript as well as the “oC” throughout the manuscript. You should add space between digit and the unit.

5. Is the rationale for adopting direct current (DC) appliances in remote buildings adequately explained? Are the limitations of alternating current (AC) electricity in supplying remote buildings effectively addressed?

6. How does the proposed use of DC appliances contribute to an efficient power system? Is the removal of inverters, power filters, insulation transformers, and complicated controllers justified? Are there any potential drawbacks or challenges associated with relying solely on DC appliances?

7. Are the results of the simulation using MATLAB/Simulink provided? Are there any performance metrics or comparisons to assess the effectiveness of the proposed DC power unit?

minor issues.

Author Response

Manuscript ID sustainability-2472070.R1

“Efficient photovoltaic unit for power delivering to stand-alone direct current buildings using artificial intelligence approach based MPP tracker"

Response to Reviewer 3

The authors would like to thank reviewer-3 for his/her constructive comments, which have undoubtedly improved the quality of the paper in its revision. In what follows, the reviewer’s remarks are in standard font, while the reply of the authors is in italics. Relevant changes in the paper are highlighted in GREEN.

Reviewer 3:

Comments and Suggestions for Authors

Recommendation: Minor revisions needed as noted.

  1. Is there a published article that is similar in the literature?
  • There is no similar published article in the literature.
  1. In the introduction section, it must be made apparent how the present work is innovative. The results of the investigation are not clearly explained.
  • The introduction section is improved and it included additional references that compared the system design and performance of the presented work with other previous researches as shown in Table 5 in the revised manuscript.
  1. What is the difference between your work and this paper (A new intelligent MPPT method for stand-alone photovoltaic systems operating under fast transient variations of shading patterns)?
  • Firstly, this paper (A new intelligent MPPT method for stand-alone photovoltaic systems operating under fast transient variations of shading patterns) is added as Ref. [15] in the revised manuscript. Secondly, the work of Ref. [15] proposed a new tracking loop based on fuzzy logic controller (FLC) with a scanning and storing algorithm. Ref. [15] focused on proposing global MPP tracker under partially shaded conditions (PSCs) for stand-alone photovoltaic (PV) systems.

 Ref. [15] is involved in the comparison Table 5 of the revised manuscript to compare many parameters with the presented work. Ref. [15] did not focus on power calculation for the electrical appliances of a typical building. The presented work proposed four layers ANN algorithm for MPPT functioning. In addition, the presented work demonstrates the advantages of adopting DC power delivering system instead of AC system.

  1. Carefully check the superscript and subscript as well as the “oC” throughout the manuscript. You should add space between digit and the unit.
  • The authors thank the reviewer for this comment. The comment is fully followed in the revised manuscript.
  1. Is the rationale for adopting direct current (DC) appliances in remote buildings adequately explained? Are the limitations of alternating current (AC) electricity in supplying remote buildings effectively addressed?
  • Normally, the stages of PV system that supply AC electricity included (PV array, MPP tracker through DC-DC converter and battery bank, 1ph/3ph inverter, low pass power filter, and step-up transformer).

The three stages of the PV system can be removed (inverter, filter, and transformer) when the system is used to deliver DC electricity. This stage reduction will improve the system performance, and reduce the system’s size and cost. This explanation is clearly demonstrated in the revised manuscript.

  1. How does the proposed use of DC appliances contribute to an efficient power system? Is the removal of inverters, power filters, insulation transformers, and complicated controllers justified? Are there any potential drawbacks or challenges associated with relying solely on DC appliances?
  • The efficiency of multi-stage system equals to the multiplication result of the efficiencies of all the system’s stages [29], so when the system’s stages are reduced, this will surely improve the system efficiency.
  •  The main challenge is that the normally used electrical appliances are suitable to connect to the AC grid because the usually used appliances are from the AC category, whereas the proposed solution adopts DC appliances.
  1. Are the results of the simulation using MATLAB/Simulink provided? Are there any performance metrics or comparisons to assess the effectiveness of the proposed DC power unit?
  • A detailed comparison table (Table 5) is added to the revised manuscript to demonstrate the merits of the presented work.

Comments on the Quality of English Language

minor issues.

  • Editing of the English language is completed in the revised manuscript.

Reviewer 4 Report

The study has serious shortcomings. These are listed below. These should be re-evaluated after they have been corrected.

1) Abstract section should be supported with meaningful numerical information.

2) Keywords should be updated. Scientific keywords should be selected.

3) The contribution of this paper to the literature should be listed in clear terms.

4) There are texts that do not appear in Figure 1 and Figure 2.

5) Abbreviations of repeated phrases should be spread throughout the text.

6) The study is problematic in terms of innovation. There are many studies where ANN is used. What is innovation?

7) Just using ANN is not enough. It is necessary to compare performance with different approaches. The success of the proposed method should be proven.

8) There is no discussion section.

9) The literature review is insufficient. The following articles about MPPT can be used.

-Kaya, C. B., Kaya, E., & Gokkus, G. (2023). Training Neuro-Fuzzy by Using Meta-Heuristic Algorithms for MPPT. COMPUTER SYSTEMS SCIENCE AND ENGINEERING45(1), 69-84.

-Messalti, S., Harrag, A., & Loukriz, A. (2017). A new variable step size neural networks MPPT controller: Review, simulation and hardware implementation. Renewable and Sustainable Energy Reviews68, 221-233.

10) The training and testing process of ANN should be mentioned in detail.

Author Response

Manuscript ID sustainability-2472070.R1

“Efficient photovoltaic unit for power delivering to stand-alone direct current buildings using artificial intelligence approach based MPP tracker"

Response to Reviewer 4

The authors would like to thank reviewer-4 for his/her constructive comments, which have undoubtedly improved the quality of the paper in its revision. In what follows, the reviewer’s remarks are in standard font, while the reply of the authors is in italics. Relevant changes in the paper are highlighted in GREEN.

Reviewer 4:

Comments and Suggestions for Authors

The study has serious shortcomings. These are listed below. These should be re-evaluated after they have been corrected.

1)   Abstract section should be supported with meaningful numerical information.      

     * The abstract is supported by numerical information in the revised manuscript

2) Keywords should be updated. Scientific keywords should be selected.

  *  Keywords are updated, and scientific keywords are selected in the revised manuscript.

3) The contribution of this paper to the literature should be listed in clear terms.

  • The contribution of the presented work is demonstrated in improved explanation in the abstract, introduction, and the additional comparison table (Table 5) of the revised manuscript.

4) There are texts that do not appear in Figure 1 and Figure 2.

  • Figure 1 and Figure 2 are fully reconstructed in the revised manuscript.

5) Abbreviations of repeated phrases should be spread throughout the text.

  • The abbreviations of repeated phrases are spread in an improved way in the revised manuscript.

6) The study is problematic in terms of innovation. There are many studies where ANN is used. What is innovation?

  • The main contribution of this paper is proposing an integrated design of a DC unit of 11 kW·h PV system for stand-alone building applications that eliminates three stages compared to AC unit and improves the system performance and increases the system’s merits. The introduced study includes PV array calculation based on PV module of 220 W with an intelligent algorithm to guarantee a fast and accurate MPP tracking for continuously harvesting a maximum power from the incident sunlight.

7) Just using ANN is not enough. It is necessary to compare performance with different approaches. The success of the proposed method should be proven.

  • System stages, performance, and specifications including MPPT algorithm, are all compared in comparison Table 5 of the revised manuscript. Many parameters including power calculation of a typical building, the advantages of the DC power delivering system instead of AC system, as well as MPPT algorithm schemes including ANN.

8) There is no discussion section.

  • Discussing the merits of the presented work with a detailed comparison table are added to the end of Section 8 of the revised manuscript.

9) The literature review is insufficient. The following articles about MPPT can be used.

-Kaya, C. B., Kaya, E., & Gokkus, G. (2023). Training Neuro-Fuzzy by Using Meta-Heuristic Algorithms for MPPT. COMPUTER SYSTEMS SCIENCE AND ENGINEERING, 45(1), 69-84.

-Messalti, S., Harrag, A., & Loukriz, A. (2017). A new variable step size neural networks MPPT controller: Review, simulation and hardware implementation. Renewable and Sustainable Energy Reviews, 68, 221-233.

  • The above researches are discussed in the revised manuscript and in the comparison table (Table 5) and added to the reference list ([24], and [25]).

  • Furthermore, additional studies from 2022 and 2023 are included in the introduction section of the revised manuscript, and added to the reference list ([17]-[22]).

10) The training and testing process of ANN should be mentioned in detail.

  • The training and testing process of the proposed ANN are shown in Figure 18 of the revised manuscript.

Round 2

Reviewer 4 Report

It has reached an acceptable level.